# Efficacy and Safety of Naldemedine for Patients with Cancer with Opioid-Induced Constipation in Clinical Practice: A Real-World Retrospective Study

**DOI:** 10.3390/jcm11092672

**Published:** 2022-05-09

**Authors:** Hiromi Nishiba, Hisao Imai, Yukiyoshi Fujita, Eriko Hiruta, Takashi Masuno, Shigeki Yamazaki, Hajime Tanaka, Teruhiko Kamiya, Masako Ito, Satoshi Takei, Masato Matsuura, Junnosuke Mogi, Koichi Minato, Kyoko Obayashi

**Affiliations:** 1Division of Pharmacy, Japan Community Health Care Organization (JCHO) Gunma Chuo Hospital, 1-7-13, Kouun, Maebashi 371-0025, Gunma, Japan; nishiba-hiromi@gunma.jcho.go.jp; 2Graduate School of Pharmaceutical Sciences, Takasaki University of Health and Welfare, 37-1, Nakaorui, Takasaki 370-0033, Gunma, Japan; obayashi@takasaki-u.ac.jp; 3Department of Respiratory Medicine, Comprehensive Cancer Center, International Medical Center, Saitama Medical University, 1397-1, Yamane, Hidaka 350-1298, Saitama, Japan; 4Division of Respiratory Medicine, Gunma Prefectural Cancer Center, 617-1, Takahayashi-nishi, Ota 373-8550, Gunma, Japan; kminato@gunma-cc.jp; 5Division of Pharmacy, Gunma Prefectural Cancer Center, 617-1, Takahayashi-nishi, Ota 373-8550, Gunma, Japan; yfujita@gunma-cc.jp (Y.F.); e-hiruta@gunma-cc.jp (E.H.); 6Division of Pharmacy, Fujioka General Hospital, 813-1, Nakagurisu, Fujioka 375-8503, Gunma, Japan; ta-masuno@fujioka-hosp.or.jp; 7Division of Pharmacy, Kiryu Kosei General Hospital, 6-3, Orihime, Kiryu 376-0024, Gunma, Japan; yakuzaibu1@kosei-hospital.kiryu.gunma.jp; 8Division of Pharmacy, Haramachi Red Cross Hospital, 698, Haramachi, Higashiagatsuma-machi, Agatsuma-gun 377-0882, Gunma, Japan; yakuzai@haramachi.jrc.or.jp; 9Department of Pharmacy, Tatebayashi Kosei General Hospital, 262-1, Narushima, Tatebayashi 374-8533, Gunma, Japan; t-kamiya@tatebayashikoseibyoin.jp; 10Division of Pharmacy, Ota Memorial Hospital, 455-1, Oshima, Ota 373-8585, Gunma, Japan; m.itou.03464@ota-hosp.or.jp; 11Division of Pharmacy, Tone Central Hospital, 910-1, Numasu, Numata 378-0012, Gunma, Japan; yak-tone03@tonehoken.or.jp; 12Division of Pharmacy, Gunma Saiseikai Maebashi Hospital, 564-1, Kamishinden, Maebashi 371-0821, Gunma, Japan; ma-matsuura@maebashi.saiseikai.or.jp; 13Division of Pharmacy, Hidaka Hospital, 886, Nakao, Takasaki 370-0001, Gunma, Japan; yakuzai2147@hidaka-kai.com

**Keywords:** clinical practice, efficacy, naldemedine, opioid-induced constipation

## Abstract

The efficacy and safety of naldemedine for opioid-induced constipation in patients with cancer has not been investigated in clinical practice. We conducted a multicenter, retrospective study to assess the effects of naldemedine among 10 Japanese institutions between June 2017 and August 2019. We evaluated the number of defecations 7 days before and after naldemedine administration. A total of 149 patients (89 male) with a median age of 72 years (range, 38–96) were included. The performance status was 0–1, 2, and ≥3 in 40, 38, and 71 patients, respectively. The median opioid dose in oral morphine equivalents was 30 mg/day (range: 7.5–800 mg). We observed 98 responders and 51 non-responders. The median number of defecations increased significantly in the 7 days following naldemedine administration from three to six (*p* < 0.0001). Multivariate analysis revealed that an opioid dose <30 mg/day [odds ratio, 2.08; 95% confidence interval, 1.01–4.32; *p* = 0.042] was significantly correlated with the effect of naldemedine. Diarrhea was the most common adverse event (38.2%) among all grades. The efficacy and safety of naldemedine in clinical practice are comparable to those of prospective studies, suggesting that it is effective in most patients.

## 1. Introduction

Patients with cancer routinely receive opioids for pain management. Prevention of adverse events, such as drowsiness, nausea, vomiting, and constipation, is vital for the continuation of pain management with opioids. Although drowsiness, nausea, and vomiting often occur at opioid initiation and dose escalation, they often improve with tolerance [1,2,3,4]. In contrast, opioid-induced constipation (OIC), which occurs when opioids act on peripheral μ-opioid receptors, reducing intestinal activation and decreasing quality of life, does not often resolve with tolerance [5]. OIC, characterized by functional constipation, is defined as a change of defecation frequency in bowel habits or patterns of defecation after opioid therapy initiation [3]. When diagnosed using the international Rome IV diagnostic criteria of OIC, the incidence of OIC was reported to be 56% in a survey in Japan [6].

Recently, the peripheral μ-opioid receptor antagonist naldemedine has become available for OIC. Naldemedine is a peripheral μ-opioid receptor antagonist (conventional PAMORA) of δ- and κ-opioid receptors to a similar degree [7,8]. This drug binds to μ-opioid receptors in the gastrointestinal tract and reduces OIC [9]. Moreover, the efficacy and safety of naldemedine in patients with cancer have been reported in randomized phase III trials such as COMPOSE-4 and COMPOSE-5 [10,11]. However, these phase III trials only included patients with an Eastern Cooperative Oncology Group-performance status (ECOG-PS) ≤ 2 who received a stable daily dose of opioids for 2 weeks before screening. Conversely, the efficacy and safety for patients with a poor PS (≥3) and those who started naldemedine early (<2 weeks after the start of opioid therapy) have not yet been fully evaluated in prospective clinical trials.

The results of a post-marketing surveillance evaluation of the safety and efficacy of naldemedine in clinical practice were recently published. It was found that naldemedine was well-tolerated and effective in patients with various backgrounds and treatment factors, such as age, sex, PS, presence or absence of organ damage, complications, and use of laxatives [12]. Moreover, the National Comprehensive Cancer Network guidelines recommend that PAMORAs, such as naldemedine, be considered when the response to laxative therapy is insufficient for managing OIC. However, naldemedine is often used in clinical practice immediately after initiating opioids or in combination with other laxatives [13]. Furthermore, previous studies in patients with OIC demonstrated that no apparent baseline patient characteristics affected the efficacy or safety of naldemedine [14,15]. Finally, naldemedine use in clinical practice for treating OIC may not be sufficiently effective.

We previously surveyed naldemedine use in clinical practice [13]; however, its efficacy has not been fully investigated. Moreover, the report stated that real-world practice for OIC management guidelines was not followed. Therefore, we conducted a multicenter, retrospective study among patients with cancer using opioids, focusing on the efficacy and safety of naldemedine in clinical practice. Our study included patients with a poor PS and elderly patients who did not meet the eligibility criteria for clinical trials.

## 2. Materials and Methods

### 2.1. Patients

This retrospective study assessed the clinical effects of naldemedine administration in patients with cancer receiving opioid therapy upon admission to 10 Japanese institutions between June 2017 and August 2019. Eligible patients were identified using the electronic medical charts and pharmacy databases. A total of 389 patients with cancer who received naldemedine for the first time in conjunction with opioid medication during hospitalization were identified. Of those, 158 patients who could be observed for at least 7 days before and after the start of naldemedine administration were identified. Seven patients with missing data and two with colostomy whose defecation frequency was difficult to assess were excluded. Finally, 149 patients were included in the analysis (Figure 1). The eligibility criteria were as follows: (i) pathologically diagnosed malignancy; (ii) naldemedine initiated during hospitalization; (iii) naldemedine administered in conjunction with opioids; (iv) hospitalization for at least 7 days before and after naldemedine administration; and (v) frequency of defecation was measured and recorded in the medical record. Patients with colostomies were excluded because of difficulty in counting the number of bowel movements. We reviewed the patients’ charts to collect data regarding baseline characteristics and responses to naldemedine. The study design was approved by the institutional review board of each participating institution. The requirement for informed consent was waived due to the retrospective nature of the study. However, the opportunity to refuse participation through the opt-out method was guaranteed.

### 2.2. Treatment

The patients included in the current study had not previously received naldemedine. They received 0.2 mg of naldemedine orally once a day with opioids, which was continued until termination was judged necessary by an attending physician, unacceptable toxicity, or withdrawal of consent.

### 2.3. Assessment of Treatment Efficacy

The number of defecations (times/week) was evaluated for 7 days before and after naldemedine administration. A responder was defined as a patient with three or more defecations/week in the first 7 days after naldemedine initiation and an increase of one or more defecations/week from baseline (the number of defecations during the week before naldemedine initiation). Adverse events were graded using the Common Terminology Criteria for Adverse Events version 5.0 (CTCAE v5.0).

We also investigated the possible association between body mass index (BMI) and the efficacy of naldemedine using a BMI cutoff of 22.0, which is the ideal BMI in the Japanese population (high BMI: ≥22.0; low BMI: <22.0) [16]. BMI was determined upon treatment initiation and was defined as the weight (kg) divided by height (m) squared.

### 2.4. Statistical Analyses

The Wilcoxon signed-rank test was used to assess normality and equal variances and test for correspondence between the two groups. Multivariate ordered logistic regression analysis was used to identify factors that predicted efficacy, and the results are described as odds ratios (ORs) and 95% confidence intervals (CIs). Differences were considered statistically significant at a two-tailed *p*-value of ≤0.05. All analyses were conducted using JMP software for Windows (version 11.0; SAS Institute, Cary, NC, USA).

## 3. Results

### 3.1. Patient Characteristics

Of the 149 patients included in the current analysis, 117 died during the study. The median time from naldemedine initiation to death was 35 days (range, 7–790 days). Patient characteristics are shown in Table 1. Briefly, 89 patients were male, the median age was 72 years (range, 38–96 years), and 60 (40.2%) were ≥75 years. According to the ECOG criteria, 40 patients (26.8%) had a PS of 0–1, 38 (25.5%) had a PS of 2, and 71 (47.6%) had a poor PS of 3–4. The most common type of malignancy in this cohort was thoracic cancer in 37 patients (24.8%), followed by liver, biliary tract, and pancreatic cancer (hepatobiliary tumors) in 34 patients (22.8%), and gastrointestinal cancer in 33 patients (22.1%).

The use of opioids, laxatives, and antiemetic agents is summarized in Table 2. The median regular opioid dose in oral morphine equivalents was 30 mg/day (range: 7.5–800 mg). Oxycodone was the most commonly used opioid, with 87 patients (58.3%) receiving 10 mg of oxycodone (15 mg of morphine equivalent). Moreover, 118 patients (79.2%) received concomitant laxatives, among whom 93 (62.4%) received magnesium oxide. Furthermore, 37 patients (24.8%) initiated naldemedine treatment within 7 days of opioid initiation, and 15 patients (10%) started both drugs on the same day. Regarding antiemetic agents, 102 patients (68.5%) received concomitant antiemetic agents, among whom 19 (12.8%) received metoclopramide.

### 3.2. Treatment Efficacy and Safety

As shown in Figure 2, 98 (65.7%; 95% CI, 58.1–73.3) and 51 patients were responders and non-responders, respectively.

Next, the frequency of defecation was compared for 1 week before and after naldemedine administration. Then, the change in the frequency of defecation was evaluated before and after treatment in the following groups: all patients, only those who defecated less than three times/week before naldemedine administration, those who received <30 mg/day of morphine equivalent, and those who received ≥30 mg/day of morphine equivalent (the median regular opioid dose of oral morphine equivalent was 30 mg/day, with 30 mg/day being the cutoff) (Figure 3). The median number of defecations in the 7 days before and after naldemedine administration in the overall population was 3 (range, 0–15) and 6 (range, 0–49), respectively. Thus, the number of defecations increased significantly after naldemedine administration (Wilcoxon signed-rank test, *p* < 0.0001; Figure 3a).

Moreover, the frequency of defecation during the 7 days was compared before and after naldemedine administration in specific populations. Among patients who had <3 defecations in the week before naldemedine administration (N = 66), the median number of defecations during the 7 days before and after naldemedine administration was 1 (range, 0–2) and 5 (range, 0–39), respectively. Thus, the number of defecations increased significantly after naldemedine administration (Wilcoxon signed-rank test, *p* < 0.0001; Figure 3b).

The frequency of defecation in the 7 days before and after the start of naldemedine administration was further compared according to opioid dose (morphine equivalent dose of <30 mg/day and ≥30 mg/day). Among patients who received a morphine equivalent of <30 mg/day (N = 60), the median number of defecations in the 7 days before and after naldemedine administration was 3 (range, 0–15) and 6 (range, 0–21), respectively. Thus, the number of defecations increased significantly after naldemedine administration (Wilcoxon signed-rank test, *p* = 0.0005; Figure 3c). When the evaluation was limited to only patients who received ≥30 mg/day of morphine equivalent (N = 89), the median number of defecations in the 7 days before and after naldemedine administration was 3 (range, 0–14) and 7 (range, 0–49), respectively. Thus, the number of defecations increased significantly in this group as well after naldemedine administration (Wilcoxon signed-rank test, *p* < 0.0001; Figure 3d).

Adverse events judged to be causally related to naldemedine administration are shown in Table 3. Diarrhea was the most common adverse event among all grades, occurring in 57 patients (38.2%), of which 52 (88.1%) were grade 1–2. None of the patients experienced grade 4 or higher adverse events based on our evaluation using CTCAE v5.0.

### 3.3. Clinical Factors Influencing Treatment Response

Next, the relationship between the efficacy of naldemedine and various clinical factors was analyzed. Multivariate logistic regression analysis was performed for responders (Table 4). A regular opioid dose of morphine equivalent <30 mg/day [OR, 2.08; 95% CI, 1.01–4.32; *p* = 0.042] was identified as a significant factor related to the effect of naldemedine. In contrast, sex (male/female), age (<75/≥75), PS (0–2/≥3), BMI (<22/≥22), use of concomitant laxatives before starting naldemedine, and history of chemotherapy within 21 days prior to naldemedine administration did not show statistically significant differences in the effect of naldemedine.

## 4. Discussion

In this study, we evaluated the frequency of defecation and adverse events in patients with cancer receiving opioids in clinical practice hospitalized for at least 7 days before and after the start of naldemedine. In addition, various clinical factors related to efficacy were examined using multivariate logistic regression analysis.

Of the patients included in the study, 65.7% were responders. In other words, 98 patients with ≥3 defecations in the first 7 days after naldemedine initiation had an increase of ≥1 defecation from baseline. This value was comparable to the responder rate of the COMPOSE-4 trial (71%) and that of naloxegol (73%), the same PAMORA [10,17]. In addition, among all patients, for only those who defecated less than three times in the week before naldemedine administration, those who received <30 mg/day of morphine equivalent, and those who received ≥30 mg/day of morphine equivalent, the frequency of defecation consistently increased after naldemedine administration. In particular, a statistically significant improvement in the frequency of bowel movements was observed in the patients’ group judged to be constipated, with fewer than three bowel movements in the week prior to naldemedine administration, indicating that the drug is likely to be useful. Moreover, in multivariate logistic regression analysis, a morphine-equivalent opioid dose of <30 mg/day was identified as a significant factor associated with the efficacy of naldemedine. Naldemedine will generally be effective in the initial stage of opioid administration due to its pharmacological properties because a low morphine equivalent dose is often administered in the initial stage of opioid treatment. Although the concomitant use of naldemedine with other laxatives may affect the efficacy of naldemedine, the multivariate logistic regression analysis results showed that the use of other laxatives prior to naldemedine administration had no significant effect on efficacy. Previous reports have not identified any baseline patient characteristics that affect efficacy in patients with OIC [14,15]. Our results were consistent with these findings, as sex, age, PS, and BMI did not significantly affect efficacy in the current study.

In the COMPOSE-4 and COMPOSE-5 randomized phase III trials of naldemedine in patients with cancer presenting with OIC, those with PS ≥ 3 were excluded [10]. Meanwhile, in the present study, 47.6% of the patients had a PS ≥ 3. This indicates that most patients receiving naldemedine in clinical practice are not those whose efficacy and safety have been demonstrated in prospective clinical trials. Moreover, although elderly patients are generally excluded from prospective clinical trials, COMPOSE-4 and -5 included patients ≥ 20 years, and no upper age limit was specified [10]. A subgroup analysis of a phase III trial reported that naldemedine was generally effective and well-tolerated in patients with chronic non-cancer pain ≥ 65 years [18]. Consistent with this report, our study found no significant difference in efficacy between patients older than 75 and those younger than 75 years of age, suggesting that naldemedine can be used to treat the elderly. A recent report on post-marketing surveillance of naldemedine evaluated its efficacy in clinical practice and reported that naldemedine was effective in patients with various background factors [12]. These results suggest that the effects of naldemedine are consistent regardless of patients’ sex, age, PS, and BMI.

Regarding safety, the most common adverse events of naldemedine were diarrhea (19.6–39.7%) and abdominal pain (1.7%) in Japanese prospective clinical trials of patients with cancer with OIC [10,19]. In the present patient population, the incidence of diarrhea and abdominal pain was consistent with these clinical trials at 57/149 (38.2%) and 4/149 (2.6%), respectively. Although our cohort included patients with a PS ≥ 3 and the elderly (aged 75 years or older), no serious adverse events, including treatment-related death, were observed except for five cases (3.3%) of grade 3 diarrhea, suggesting that naldemedine can be administered safely. In the post-marketing surveillance of naldemedine [12], the incidence of adverse drug reactions was reported to be 11.3%, of which diarrhea occurred in 9% of cases, which is less frequent than the results of previous prospective studies. This inconsistency could be due to bias from the post-marketing survey and a lack of accurate counting by medical staff.

Our study had several limitations. First, this was a retrospective analysis, which might have reduced the validity of the data gathered. Consequently, objective assessments, such as the Bristol stool form scale [20], bowel function index [21], and defecation diary, were not available for this study. Second, although naldemedine is administered to many outpatients in clinical practice, it was not possible to evaluate the efficacy of naldemedine based on spontaneous bowel movements, so the data were limited to inpatients. Inpatient data were considered more reliable than outpatient data because they were evaluated from multiple observations by different healthcare providers, such as physicians, nurses, and pharmacists. However, considering that this analysis was limited to inpatients that needed to be hospitalized for some reason, it is clinically meaningful that patients were hospitalized for at least 7 days before and after starting naldemedine treatment and were evaluated by health care professionals. Third, the decision to start or discontinue naldemedine administration was left to the discretion of each physician. Additionally, patients were undergoing various treatments for cancer, and the use of opioids and concomitant medications, such as laxatives, was not standardized due to the study’s retrospective nature. Although this study must be interpreted considering the above limitations, we believe that the results reflect clinical practice and are worth reporting. Fourth, assessment of treatment efficacy in terms of the number of bowel movements is not entirely consistent with evaluating complete spontaneous bowel movements as the primary treatment outcome.

In conclusion, the efficacy and safety of naldemedine in real-world clinical practice, where it is often administered to patients with poor PS and the elderly, are comparable to those of prospective studies, suggesting that it is effective in almost all patients, although the opioid dose may affect its efficacy. However, this was a retrospective study, and further verification in clinical practice is necessary.

## Figures and Tables

**Figure 1 jcm-11-02672-f001:**
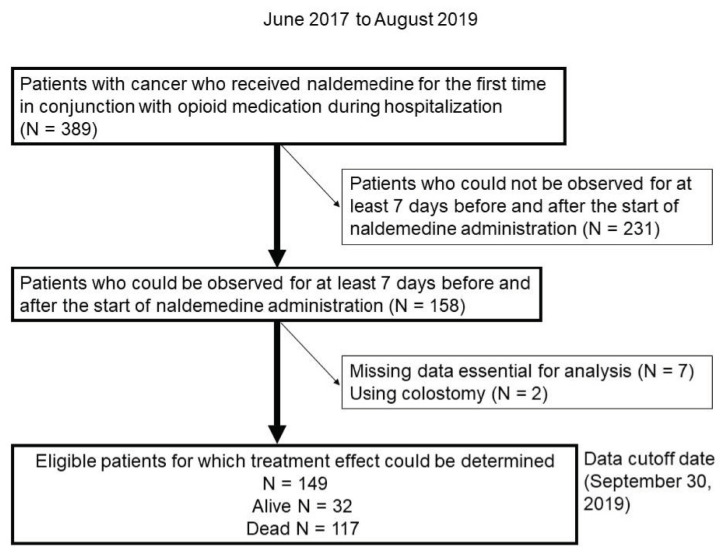
Flow chart showing patient selection. The patients received naldemedine upon first administration between June 2017 and August 2019.

**Figure 2 jcm-11-02672-f002:**
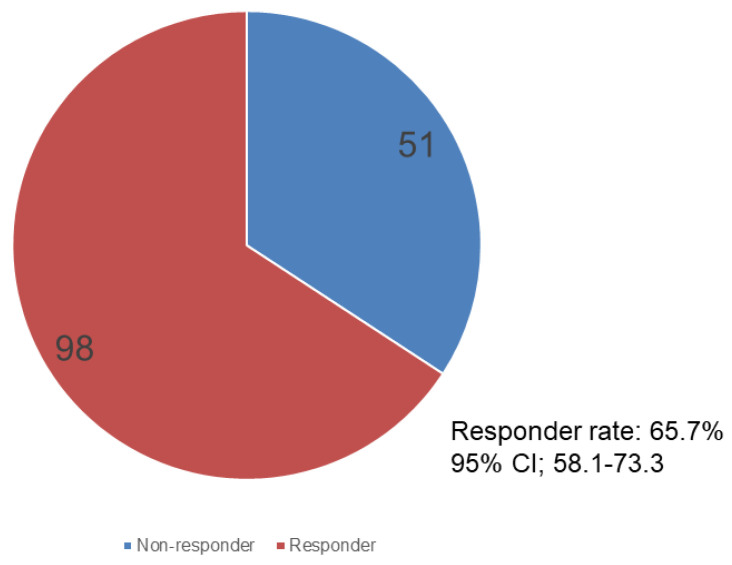
Pie chart showing responders and non-responders after naldemedine administration. Responder rate: 65.7%, 95% CI, 58.1–73.3%.

**Figure 3 jcm-11-02672-f003:**
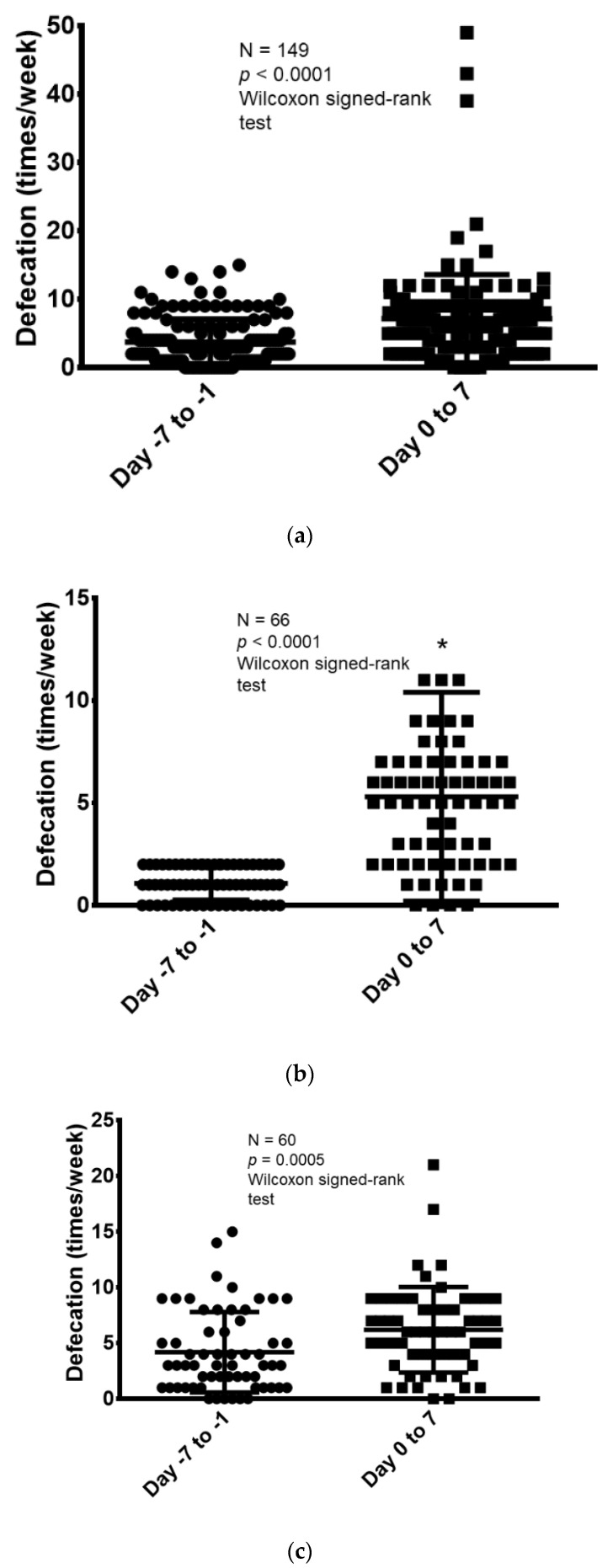
Comparison of defecation frequency before and after 7 days of naldemedine administration. (**a**) Comparison of the frequency of defecation before and after naldemedine administration in all patients (N = 149). (**b**) Comparison of defecation frequency before and after naldemedine administration among patients with defecation frequency < 3 times in the week before naldemedine administration (N = 66). * One patient data point is outside the axis limits. (**c**) Comparison of defecation frequency before and after naldemedine administration among patients who received <30 mg/day of morphine equivalent (N = 60). (**d**) Comparison of defecation frequency before and after naldemedine administration among patients who received ≥30 mg/day of morphine equivalent (N = 89). ** Three patient data points are outside the axis limits.

**Table 1 jcm-11-02672-t001:** Baseline patient characteristics.

Characteristic	N = 149 (%)
Sex	
Male	89 (59.7)
Female	60 (40.3)
Median age at treatment (years) (range)	72 (38–96)
Performance Status (PS)	
0/1/2/3/4	15/25/38/50/21 (10.1)/(16.8)/(25.5)/(33.6)/(14.1)
Primary tumor	
Thoracic cancer	37 (24.9)
Liver, biliary, and pancreatic cancer	34 (22.8)
Gastrointestinal cancer	33 (22.1)
Urinary tract, renal cell, and prostate cancer	17 (11.4)
Hematologic cancer	10 (6.7)
Gynecological cancer	8 (5.4)
Head and neck cancer	4 (2.7)
Breast cancer	4 (2.7)
Others	2 (1.3)
Therapy before and during naldemedine administration *, **	
Chemotherapy	40 (26.8)
Radiotherapy	19 (12.6)
Chemoradiotherapy	3 (2.0)
Surgery	1 (0.6)
Best supportive care alone	95 (63.8)
Central nervous system metastases	
Yes	14 (9.4)
No	135 (90.6)
Cancerous peritonitis	
Yes	19 (12.8)
No	130 (87.2)
Gastrointestinal obstruction	
Yes	0 (0)
No	149 (100)
History of abdominal surgery before starting naldemedine	
Yes	52 (34.9)
No	97 (65.1)
History of radiation to the abdomen and pelvic region before starting naldemedine	
Yes	22 (14.8)
No	127 (85.2)
Presence of diabetes mellitus	
Yes	21 (14.1)
No	128 (85.9)
Body mass index (BMI) (kg/m^2^)	
<22/≥22	106/43 (71.1)/(28.9)
Median BMI (range)	20.4 (13.7–34.8)
Discontinuation within 7 days	
Yes	19 (12.8)
No	130 (87.2)
Use of laxatives before starting naldemedine administration	
Yes	123 (82.6)
No	26 (17.4)
Use of laxatives after starting naldemedine administration	
Yes	118 (79.2)
No	31 (20.8)
Regular use of antiemetic medication after initiation of naldemedine	
Yes	44 (29.5)
No or unknown	105 (70.5)
Irregular use of antiemetic agents after starting naldemedine	
Yes	25 (16.8)
No or unknown	124 (83.2)
Survival status at data-cutoff date	
Dead	117 (78.5)
Alive	32 (21.5)
Period from naldemedine initiation to death	
Median period (days) (range)	35 (7–790)

* Within three weeks before starting naldemedine administration; ** Total number of patients.

**Table 2 jcm-11-02672-t002:** Administration of opioids, laxatives, and antiemetic agents.

Parameter	N = 149 (%)
Daily dose of opioids	
<20 mg	52 (34.9)
20–49	53 (35.6)
50–99	19 (12.7)
≥100	25 (16.8)
Regular use of opioids	
Oxycodone	87 (58.4)
Morphine	15 (10.0)
Fentanyl	32 (21.5)
Hydromorphone	11 (7.4)
Others	3 (2.0)
No regular use	1 (0.7)
Days from first opioid administration to initial naldemedine use	
<4	22 (14.8)
4–7	15 (10.0)
8–29	69 (46.3)
30–99	26 (17.5)
≥100	17 (11.4)
Concomitant laxatives **	
Magnesium oxide	93 (62.4)
Sennoside	37 (24.8)
Bisacodyl	15 (10.1)
Lubiprostone	22 (14.8)
Sodium picosulfate hydrate	15 (10.1)
Sodium bicarbonate, sodium dihydrogen phosphate anhydrous suppository	10 (6.8)
Others	9 (0.6)
Concomitant antiemetic (regular and abbreviated use) **	
Metoclopramide	22 (14.8)
Domperidone	12 (0.8)
Prochlorperazine	20 (13.4)
Olanzapine	11 (7.4)
Others	4 (2.7)
No use	31 (20.8)

** Total number of patients.

**Table 3 jcm-11-02672-t003:** Adverse events during naldemedine administration.

Adverse Events *	Grade 1	Grade 2	Grade 3	Grade 4
Diarrhea	41	11	5	0
Abdominal pain	3	1	0	-
Nausea	9	4	1	-
Anorexia	16	4	1	0
Vomiting	2	2	1	0
Fatigue	14	1	-	-

* Adverse events were graded using the Common Terminology Criteria for Adverse Events (CTCAE v5.0) version 5.0.

**Table 4 jcm-11-02672-t004:** Multivariate logistic regression analysis of responders to naldemedine.

Variables	OR	95% CI	*p*-Value
Sex			
Male/female	0.92	0.44–1.92	0.82
Age			
<75/≥75	1.35	0.63–2.956	0.42
PS			
0–2/≥3	0.96	0.46–1.98	0.91
BMI			
<22/≥22	0.77	0.35–1.69	0.51
Use of concomitant laxatives before starting naldemedine			
Yes/no	1.58	0.59–4.76	0.37
Regular dose of opioids (morphine equivalent)			
<30/≥30	2.08	1.01–4.32	**0.042**
History of chemotherapy within 21 days prior to naldemedine administration			
Yes/no	0.92	0.40–2.05	0.85

OR, odds ratio; CI, confidence interval; PS, performance status; BMI, body mass index. Values in bold type are significant (*p* < 0.05).

## Data Availability

The data presented in this study are available on request from the corresponding author. The data are not publicly available.

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
