# Peer review of "Efficacy and Safety of Naldemedine for Patients with Cancer with Opioid-Induced Constipation in Clinical Practice: A Real-World Retrospective Study"

_jcm, 2022, doi:10.3390/jcm11092672_

Round 1
Reviewer 1 Report
I read with interest the MS "Efficacy and safety of naldemedine for opioid-induced constipation in clinical practice: A retrospective study" by Nishiba H and coworkers. Opioid induced constipation (OIC) is a disabling disorder and any effort to improve its management is welcomed. However, I have some major issue about the MS, namely: A) the title needs refinement: it should be specified that the study is limited to oncology patients and refers to real world experience for OIC management guidelines were not followed. PAMORAs should be prescribed for refractory constipation. Moreover, PAMORAs should not be prescribed when peritoneal adhesions are likely as it happen in cancer peritonitis and/or in patients managed by radiotherapy, B) introduction is to be improved: the Authors should explicite the decision of not following the guidelines according to their previous real world experience, C) Ethical problems: I understand it is a retrospective study, but some documentation of the quoted opt-out procedure is to be presented, D) assessment of treatment efficacy by number of bowel movements, this is a major limitation which is to be addedd to the MS for the actual common use of "complete spontaneous bowel movements" as main treatment outcome, E) the subgroup of patients at risk of bowel obstruction (cancer peritonitis and previous radiotherapy any time) is to be analysed separetely in terms of both efficacy and safety for the actual discouraging of the use of PAMORAs in these patients. The discussion needs to be changed accordingly either or not supporting guidelines. Minor point. use of word PAROMA instead of PAMORA on line 56. No additional suggestions on this side.
Author Response
Thank you for taking the time out of your busy schedule to include your very important points and review. We will respond to you as follows.
Review Report (Reviewer 1)
Comments and Suggestions for Authors
I read with interest the MS "Efficacy and safety of naldemedine for opioid-induced constipation in clinical practice: A retrospective study" by Nishiba H and coworkers. Opioid induced constipation (OIC) is a disabling disorder and any effort to improve its management is welcomed. However, I have some major issue about the MS, namely: A) the title needs refinement: it should be specified that the study is limited to oncology patients and refers to real world experience for OIC management guidelines were not followed. PAMORAs should be prescribed for refractory constipation. Moreover, PAMORAs should not be prescribed when peritoneal adhesions are likely as it happen in cancer peritonitis and/or in patients managed by radiotherapy, B) introduction is to be improved: the Authors should explicite the decision of not following the guidelines according to their previous real world experience, C) Ethical problems: I understand it is a retrospective study, but some documentation of the quoted opt-out procedure is to be presented, D) assessment of treatment efficacy by number of bowel movements, this is a major limitation which is to be addedd to the MS for the actual common use of "complete spontaneous bowel movements" as main treatment outcome, E) the subgroup of patients at risk of bowel obstruction (cancer peritonitis and previous radiotherapy any time) is to be analysed separetely in terms of both efficacy and safety for the actual discouraging of the use of PAMORAs in these patients. The discussion needs to be changed accordingly either or not supporting guidelines. Minor point. use of word PAROMA instead of PAMORA on line 56. No additional suggestions on this side.
- A) the title needs refinement: it should be specified that the study is limited to oncology patients and refers to real world experience for OIC management guidelines were not followed. PAMORAs should be prescribed for refractory constipation. Moreover, PAMORAs should not be prescribed when peritoneal adhesions are likely as it happen in cancer peritonitis and/or in patients managed by radiotherapy
Response: Thank you for your very constructive comments.
The title has been completely and substantially revised as follows: “Efficacy and safety of naldemedine for patients with cancer with opioid-induced constipation in clinical practice: A real-world retrospective study”
- B) introduction is to be improved: the Authors should explicite the decision of not following the guidelines according to their previous real world experience
Response: Thank you for your comment. We have added the following sentence to the text as you suggested (Page 2, lines 80-81): “Moreover, the report described that real-world experience for OIC management guidelines was not followed. “
- C) Ethical problems: I understand it is a retrospective study, but some documentation of the quoted opt-out procedure is to be presented
Response: Thank you for your comment. We have added the following sentence to the text as you suggested (Page 12, lines 339-340): “However, the opportunity to refuse participation through the opt-out method was guaranteed.”
- D) assessment of treatment efficacy by number of bowel movements, this is a major limitation which is to be addedd to the MS for the actual common use of "complete spontaneous bowel movements" as main treatment outcome
Response: Thank you for your very insightful comments. We have added the following sentence to the text as you suggested (Page 12, lines 313-315): “Fourth, assessment of treatment efficacy in terms of the number of bowel movements is not entirely consistent with evaluating complete spontaneous bowel movements as the primary treatment outcome. “
- E) the subgroup of patients at risk of bowel obstruction (cancer peritonitis and previous radiotherapy any time) is to be analysed separetely in terms of both efficacy and safety for the actual discouraging of the use of PAMORAs in these patients. The discussion needs to be changed accordingly either or not supporting guidelines.
Response: Thank you very much for your insightful remarks. As you point out, cancerous peritonitis and history of abdominal irradiation are important matters. However, we do not know from the data we have whether cancerous peritonitis is only seen on imaging or whether there are symptoms, and we do not know the extent of abdominal radiotherapy, such as whether the intestinal tract is involved, so there are limitations to the analysis. We are also now analyzing all the cases, but we plan to conduct a detailed study focusing on gastrointestinal cancers as a secondary analysis. At that time, we plan to focus on cancer peritonitis and a history of abdominal radiotherapy.
Minor point. use of word PAROMA instead of PAMORA on line 56. No additional suggestions on this side.
Response: Thank you for your suggestion, we have revised the term from PAROMA to PAMORA.
Reviewer 2 Report
The authors retrospectively studied the efficacy, adverse events, and factors associated with the response of naldemedine in patients with cancer receiving opioids in clinical practice. This study extended the clinical trial data on special subgroups such as patients with low-performance status and the elderly.
- About 25% of patients started naldemedine within 7 days after opioid and 10% of patients started both on the same day. So, the inclusion criteria were not clear as about one-third of patients probably do not have opioid-induced constipation, but use naldemedine for prophylaxis and the treatment efficacy was difficult to interpret. It would be better if the authors could add more information regarding patients symptoms or laxatives usage before prescribing naldemedine in this group.
- The efficacy of naldemedine could be supported if the authors showed the laxatives dosages for 7 days before and after taking naldemedine (also other constipation symptoms, and enema or manual maneuvers). At least, it should be stable or not increase. This data can be reviewed as all patients were hospitalized.
- Table 1 can be improved by deleting Yes or No at appropriate points.
- It would be great if the authors show the adverse effects in the special group (age > 75 and performance status 3-4)
Author Response
Thank you for taking the time out of your busy schedule to include your very important points and review. We will respond to you as follows.
Review Report (Reviewer 2)
Comments and Suggestions for Authors
The authors retrospectively studied the efficacy, adverse events, and factors associated with the response of naldemedine in patients with cancer receiving opioids in clinical practice. This study extended the clinical trial data on special subgroups such as patients with low-performance status and the elderly.
- About 25% of patients started naldemedine within 7 days after opioid and 10% of patients started both on the same day. So, the inclusion criteria were not clear as about one-third of patients probably do not have opioid-induced constipation, but use naldemedine for prophylaxis and the treatment efficacy was difficult to interpret. It would be better if the authors could add more information regarding patients symptoms or laxatives usage before prescribing naldemedine in this group.
Response: Thank you for your very pertinent points. As you pointed out, we have data on the dose and type of opioid and laxative used at the time of naldemedine administration. However, we did not obtain data on the patient's symptoms and laxative use prior to naldemedine administration. We evaluated the data as a close approximation of the data obtained at the time of use. We would like to consider obtaining and analyzing such data as you indicated as part of our future analysis.
- The efficacy of naldemedine could be supported if the authors showed the laxatives dosages for 7 days before and after taking naldemedine (also other constipation symptoms, and enema or manual maneuvers). At least, it should be stable or not increase. This data can be reviewed as all patients were hospitalized.
Response: Thank you for your instruction. We presently have not obtained data on detailed laxative doses before and after naldemedine administration. It is difficult to obtain such data from multiple institutions in a short period due to the revision within 10 days of the revision deadline. We will reanalyze the data based on what you have indicated in future secondary analyses.
- Table 1 can be improved by deleting Yes or No at appropriate points.
Response: Thank you for your suggestion. We have revised Table 1.
- It would be great if the authors show the adverse effects in the special group (age > 75 and performance status 3-4)
Response: Thank you very much for your insightful remarks, which will lead to future secondary analysis. We plan to conduct a secondary analysis of the population, including cases with poor PS and elderly patients, as a future secondary analysis since the number of Tables and Figures in this study is too large.
Round 2
Reviewer 1 Report
I read with interest the revised version of this real-world retrospective, relevant study dealing with OIC managed by Naldemedine prescription. All of my queries have been addressed in full. No additional suggestions on this side.